

# Maintenance of xylem hydraulic function during winter in the woody bamboo *Phyllostachys propinqua* McClure

Yongxin Dai[1,2], Lin Wang[1,2] and Xianchong Wan[2]

[1] College of Forestry, Shanxi Agricultural University, Taigu, Shanxi, China
[2] Institute of New Forestry Technology, Chinese Academy of Forestry, Beijing, China

## ABSTRACT

**Background**. Frost is a common environmental stress for temperate plants. Xylem embolism occurs in many overwintering plants due to freeze-thaw cycles, so coping with freeze-thaw-induced embolisms is essential for the survival of temperate plants.

**Methods**. This study was conducted on *Phyllostachys propinqua* McClure, a woody bamboo species that was grown under natural frost conditions to explore its responses to winter embolisms. From autumn to the following spring, the following measurements were recorded: predawn branch and leaf embolism, branch and leaf relative water content (RWC), root pressure and soil temperature, xylem sap osmotic potential, branch and leaf electrolyte leakage (EL), branch nonstructural carbohydrate (NSC) content and leaf net photosynthetic rate.

**Results**. *P. propinqua* had a mean vessel diameter of 68.95 ±1.27 μm but did not suffer severe winter embolism, peaking around 60% in winter (January), with a distinct reduction in March when root pressure returned. Leaves had a more severe winter embolism, up to 90%. Leaf RWC was much lower in winter, and leaf EL was significantly higher than branch EL in all seasons. Root pressure remained until November when soil temperature reached 9 °C, then appeared again in March when soil temperatures increased from −6 °C (January) to 11 °C. Xylem sap osmotic potential decreased from autumn to winter, reaching a minimum in March, and then increasing again. Soluble sugar (SS) concentration increased throughout the winter, peaked in March, and then decreased.

**Conclusions**. These results suggest that (1) there is a hydraulic segmentation between the stem and leaf, which could prevent stem water loss and further embolization in winter; (2) maintenance of root pressure in early winter played an important role in reducing the effect of freeze-thaw cycles on the winter embolism; (3) the physiological process that resulted in a decrease in xylem sap osmotic potential and tissue water content, and an accumulation of SS associated with cold acclimation also aided in reducing the extent of freeze-thaw-induced embolism. All these strategies could be helpful for the maintenance of xylem hydraulic function of this bamboo species during winter.

Corresponding authors
Yongxin Dai,
daiyongxin1234@126.com
Xianchong Wan, wxc@caf.ac.cn

## INTRODUCTION

Frost is an important climatic factor that determines the geographical distribution of plant species (*Di Francescantonio et al., 2020*). In the middle to high latitudes and alpine regions, plants are confronted with frost or sub-freezing temperatures, so plants able to resist frost may be more competitive and have higher survivability. Liquids freezing within the plants may occur in two different ways: symplastic freezing, which causes cytolysis or cell death, or apoplastic freezing, which causes embolism formation in the xylem conduits, constraining long-distance water transport capacity (*Charra-Vaskou et al., 2016*). In woody plants, freezing usually occurs first in the apoplastic spaces, affecting the cytoplasm and cell membrane of living cells, but not necessarily leading to symplastic freezing, depending largely on whether the plants are winter hardened (*Pearce, 2001*). If there are then usually apoplastic freezing is not accompanied by symplastic freezing, but if not, then symplastic freezing will likely occur at the same time (*Morin et al., 2007*). Apoplastic freezing also greatly increases the risk of embolization (*Lintunen et al., 2018*). Thus, freeze-thaw-induced embolism is one of the biggest challenges for plants living in northern areas or at high altitudes.

Freeze-thaw-induced embolism has been studied in a large number of species (*Sperry & Sullivan, 1992*; *Sperry et al., 1994*; *Hacke & Sauter, 1996*; *Améglio et al., 2003*; *Mayr et al., 2007*; *Mayr & Sperry, 2010*; *Mcculloh et al., 2011*). Hydraulic dysfunction due to freeze-thaw-induced embolisms may influence plant growth and fitness, including the timing of bud break in the spring (*Wang, Ives & Lechowicz, 1992*; *Cox & Zhu, 2003*), but the mechanisms and dynamics of this type of embolism still remains unclear (*Mayr et al., 2020*). It is well known that frost (freeze-thaw cycles) and drought are the two major drivers of xylem embolism, through different formation processes (*Fernández-Pérez et al., 2018*). The formation of freeze-thaw-induced embolism is classically explained by the 'thaw-expansion hypothesis' (also called the 'bubble formation hypothesis'), which is that the bubbles dissolved in xylem sap escape during sap freezing because they are insoluble in ice. During thawing, an embolism then forms if the xylem tension is high enough to cause these bubbles to expand (*Mayr et al., 2006*; *Mayr & Sperry, 2010*), with larger conduit diameters forming larger bubbles, making it easier for embolism to form. The relationship between freeze-thaw-induced embolism risk and conduit diameter has been experimentally demonstrated (*Davis, Sperry & Hacke, 1999*; *Pittermann & Sperry, 2006*; *Schreiber et al., 2013*; *Lintunen et al., 2014*), however, conduit size alone does not explain bubble volume, and low water potential and restricted bark permeability, which restricts gas from escaping, are also factors (*Lintunen et al., 2022*). Hagen–Poiseuille's law indicates that hydraulic conductance is proportional to the fourth power of the conduit radius, so a trade-off between resistance to freeze-thaw-induced embolism (hydraulic safety) and hydraulic efficiency (the capacity of the xylem to transport water) has been well established (*Anfodillo & Olson, 2021*; *Liu et al., 2021*). Other researchers, however, have found that this trade-off was absent in some species capable of repairing winter embolisms (*Niu, Meinzer & Hao, 2017*). Thus, more experimental evidence is still needed on whether some species are better equipped to cope with winter embolisms.
Woody plants growing in seasonal or high altitude environments have evolved some strategies to cope with freeze-thaw-induced embolisms: avoidance, repair and tolerance (*Sperry et al., 1994*). In plants using the avoidance strategy, a relatively small degree of embolism occurs. As stated above, narrower conduits have a lower risk of winter embolisms, which explains the distribution of plant species from wide-vessel lianas in the tropical zone to narrow-tracheid conifers in high latitudes (*Cruiziat, Cochard & Améglio, 2002*; *Taneda & Tateno, 2005*; *Jiménez-Castillo & Lusk, 2013*). The major mechanism of embolism repair is dissolving gas into the xylem sap or diffusing gas to the outside surface of the branch (*Yang & Tyree, 1992*). Embolisms can be repaired through two typical patterns: under positive pressure or under tension. Positive xylem pressure consists of root pressure and stem pressure, the actual location and mechanism of which is still unclear (*Schenk, Jansen & Hölttä, 2021*), but is widely believed to promote seasonal embolism repair (*Sperry & Tyree, 1988*; *Ewers, Cochard & Tyree, 1997*; *Ewers et al., 2001*; *Ameglio et al., 2006*; *Niu, Meinzer & Hao, 2017*). Especially in monocotyledon species and lianas, root pressure plays an important role in embolism removal, as monocotyledons cannot grow new vessels and wide-vessel lianas are at high risk of freeze-thaw-induced embolisms (*Fisher et al., 1997*; *Tibbetts & Ewers, 2000*; *Pickard, 2003a*; *Cobb, Choat & Holbrook, 2007*). Species that are capable of producing root pressure may expand their distribution ranges (*Ewers, Cochard & Tyree, 1997*; *Jiménez-Castillo & Lusk, 2013*). However, researchers also found that successful invasion of exotic temperate liana *Celastrus orbiculatus* in Michigan was not due to root pressure, but because of different responses to environmental constraints (*Tibbetts & Ewers, 2000*). Therefore, different plants may adopt different strategies to cope with winter embolisms.

Embolism repair under tension has long been studied, but is still under debate (*Choat et al., 2019*). There is, however, agreement on the involvement of osmoregulation-driven refilling, including the activity of living cells, parenchyma, phloem and aquaporin (*Secchi, Pagliarani & Zwieniecki, 2017*; *Secchi et al., 2021*; *Klein et al., 2018*; *Trifilò et al., 2019*). Seasonal embolism repair under tension has been confirmed experimentally in a few coniferous species that could fulfill this refilling process by absorbing water (melting snow) via the needle, branch or bark (*Laur & Hacke, 2014*; *Mayr et al., 2014*; *Mayr et al., 2020*; *Mason Earles et al., 2016*). Nonstructural carbohydrates (NSC) have been the topic of recent research on embolism repair under tension, because an important step in embolism repair is transporting soluble sugars from parenchyma cells around the vessels into the embolized vessels to establish an osmotic pressure gradient, which is essential for allowing water to flow into the embolized vessels (*Klein et al., 2018*; *Trifilò et al., 2019*; *Kiorapostolou et al., 2019*). Other plants without positive xylem pressure, like ring-porous species, can tolerate a very severe winter embolism without repairing it, because they instead store energy for growing new effective vessels (*Sperry et al., 1994*; *Hacke & Sauter, 1996*; *Urli et al., 2013*).

Bamboos are rapid-growing, perennial evergreen woody herbs, which play an important role in biodiversity and forest regeneration. China has the richest bamboo resources in the world with >500 species in 39 genera (*Yang et al., 2012*). Bamboos are mainly distributed in tropical and subtropical areas, with few in temperate zones, which may be related to freeze-thaw-induced embolism risk. Bamboos have wide vessels that are 50–100 $\mu$m in

diameter (*Cochard, Ewers & Tyree, 1994*; *Wang et al., 2011*), making them more vulnerable to freeze-thaw-induced embolism. Previous studies have found that 44 μm is the critical diameter of freeze-thaw-induced embolism under a water potential of −0.5 MPa, and vessels larger than 30 μm are more vulnerable (*Davis, Sperry & Hacke, 1999*; *Mayr, Gruber & Bauer, 2003*). Bamboos are monocotyledonous plants that lack secondary growth. If an embolism occurs, these plants cannot grow new vessels to restore the water-transport function (*Wang et al., 2011*). Few studies have focused on how bamboo responds to freeze-thaw-induced embolism, but previous research indicates root pressure accounts for its resistance to water stress (*Cochard, Ewers & Tyree, 1994*; *Wang et al., 2011*; *Yang et al., 2012*). Our previous study found that ring- and diffuse-porous species growing in fields experienced severe winter embolisms, with nearly 100% loss of conductivity in ring-porous species and ∼80% loss of conductivity in diffuse-porous species (*Dai, Wang & Wan, 2020*). Bamboos theoretically undergo more severe freeze-thaw-induced embolisms compared with trees growing under the same conditions, but we found that one bamboo species, *Phyllostachys propinqua* McClure, can grow normally without experiencing severe winter embolisms.

We used *P. propinqua* plants as experimental materials to examine its strategies for surviving cold climates. Between autumn and the following spring, the following measurements were taken: degree of embolism, water content, degree of freezing damage to branch and leaf, root pressure and branch nonstructural carbohydrates (NSC). We began the study with the following hypotheses: (1) bamboo plants can produce root pressure, so we hypothesized that root pressure may play a major role in *P. propinqua*'s ability to cope with freeze-thaw-induced embolisms; (2) bamboo has 'hydraulic segmentation' or 'hydraulic fuses,' meaning branches and leaves have different vulnerability to water stress (*Yang et al., 2012*), so we hypothesized that this function may contribute to protecting branches from severe freeze-thaw-induced embolisms; (3) in temperate regions, gradual low temperatures before freezing helps plants acclimatize to the cold, initiating physiological and biochemical adjustments that lead to freezing tolerance in plants (*Arias et al., 2017*). We hypothesized that cold acclimation helps *P. propinqua* cope with freeze-thaw-induced embolisms.

## MATERIAL AND METHODS

### Plant material and site

*Phyllostachys propinqua* McClure plants were cultivated in a nursery garden located in Shanxi Agricultural University, Taigu (37° 26′N, 112° 32′E, alt. ca. 780 m), China. *P. propinqua* is a scattered woody bamboo with culms that reach 2–3 m tall and 1–2 cm diameter at the base. All *P. propinqua* plants were grown under natural conditions (growing in the fields in uncontrolled climatic conditions) and normal watering management, meaning during the growing season, plants were watered according to demand, until the soil water content reached 90–100% of the field capacity. When the top 5 cm of the soil appeared obviously dry, it was watered. Once the soil froze, it was no longer watered throughout the winter, until it thawed again in the spring. The Taigu region has a temperate continental climate with four distinct seasons, warm and rainy in the summer, and cold in

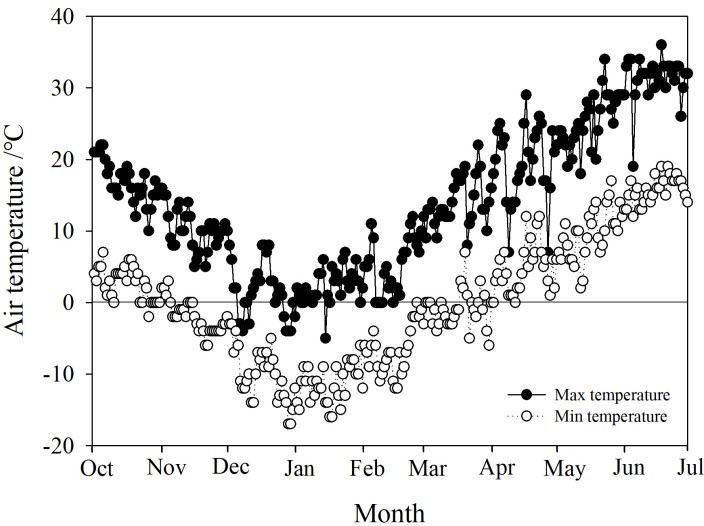

**Figure 1   Daily maximum and minimum air temperatures in the Taigu region, 2018-2019.**

the winter. Its annual average precipitation is about 463 mm, and the annual average air temperature is 5–10 °C, with a minimum temperature usually below −20 °C. Its frost-free period is about 176 days. Climate data of Taigu, including daily minimum and maximum air temperatures for the experiment periods, were downloaded from government weather stations (http://data.cma.cn/user/register/shareLevel/4.html) (Fig. 1).

The experiment was conducted from October 2018 to June 2019 with a sampling interval of one to two months. For each sampling, at least six sun-exposed branches or leaves in the middle of the plants were collected from six well-grown plants selected randomly. The sampling time at dawn was 04:30–05:30 h. Those sampling methods were applied for all following methods. Hydraulic parameters of branch and leaf were measured to assess branch and leaf hydraulic function and analyze the influence of frost on their water transport capacity. Root pressure and soil temperature were measured to analyze the relationship between the two, and to further analyze the role of root pressure in hydraulic function. Branch NSC content and xylem sap osmotic potential, leaf net photosynthetic rate and electrolyte leakage of the branch and leaf were measured to analyze degree of freezing damage to the bamboo plants, and to further analyze whether the leaf played a role in protecting the branch from severe freeze-thaw-induced embolism.

## Maximum vessel length and mean vessel diameter

For hydraulic conductivity measurement, the initial segment cut from the plant must be longer than maximum vessel length to minimize the introduction of additional air bubbles (*Sperry, Donnelly & Tyree, 1988*). Therefore, the maximum vessel length needs to be measured before cutting. The maximum vessel length of bamboo can be very large (*Cochard, Ewers & Tyree, 1994*), so for this study, the whole plant was cut at ground level. The maximum vessel length was measured using the 'air method' (*Wang et al., 2014*). Six plants were sampled and quickly brought back to the adjacent laboratory for measurement.
Both ends of the culm were trimmed via a fresh razor blade, then the apex end was immersed in tap water, and the basal end was injected with 0.15 MPa of compressed air to observe whether there were bubbles coming out from the other end. Bubbles were observed immediately, indicating that the maximum vessel length of *P. propinqua* was longer than its entire culm.

Mean vessel diameter was measured from transverse sections of the branch (*Davis, Sperry & Hacke, 1999*). Transverse sections (80–100 μm in thickness) were cut from the freshly sampled branches using a microtome (LEICA RM 2135, Wetzlar, Germany), and then immediately placed in a few drops of distilled water on a glass slide and covered with a cover slip for observation with an optical microscope (Axio Imager A1, Carl Zeiss Inc., Oberkochen, Germany) equipped with a digital camera. The photos were then analyzed using an imaging software (Axio Vision Rel 4.6) to calculate diameters. The mean diameter was determined from six branches ($n = 6$).

### Predawn water potential

Predawn water potential ($\Psi_{pd}$) was measured on six twigs using a pressure chamber (PMS1505D-EXP, Albany OR USA). Six sun-exposed twigs from six individual plants were cut at dawn (04:30–05:30 h) and measured immediately.

### Branch native embolism

The degree of embolism has traditionally been quantified by percentage loss of conductivity (PLC). In this study, predawn branch PLC was measured according to the methods outlined by *Sperry, Donnelly & Tyree (1988)*. Since the maximum vessel length of *P. propinqua* was longer than its entire culm, sampling needed to be done under water rather than in the air. Six branches from six individual plants were excised under water at dawn, and kept under water during transportation to the laboratory for immediate measurements. The hollow internodes of *P. propinqua* cannot be used directly for measurement, but the sites bearing nodes are not hollow. A branch segment with one node ensures that water flows only through vessels, not through the hollow sections, so a 3–4 cm segment with one node was cut under water from the sampled branch and used for measurements (*Cochard, Ewers & Tyree, 1994*). Both the flow and flush solution was 0.025 mol L$^{-1}$ KCl filtered through a 0.22 μm microporous membrane and then degassed. To obtain an initial conductivity ($K_i$), the KCl solution was flushed through at a lower pressure, 0.2 MPa, because higher pressures can remove the air from the embolized vessels (*Cochard, Ewers & Tyree, 1994*). The maximum conductivity ($K_{max}$) was obtained by flushing the segments with the flush solution at 0.175 MPa for 5–10 min until the conductivity no longer increased. PLC (%) was calculated as ($1 - K_i / K_{max}$) $\times 100$.

### Leaf relative embolism

Leaf hydraulic conductance ($K_{leaf}$) was obtained by examining the relaxation kinetics of leaf water potential ($\Psi_{leaf}$) in detached leaves (*Brodribb & Holbrook, 2003*; *Yang et al., 2012*). First, the pressure-volume relationship curve was drawn to obtain leaf capacitance ($C_{leaf}$) using the bench drying technique. About 10 branches, each bearing 5–6 compound leaves, were cut under water from six individual plants in autumn and immersed in distilled

water to rehydrate for 24 h until $\Psi_{leaf}$ was $\geq -0.5$ MPa. One compound leaf was detached and dried with filter paper, then immediately weighed (saturated fresh mass, $FW_1$). The leaf was air-dried at room temperature (20 °C) to desiccate for several minutes. $\Psi_{leaf}$ was then measured followed by sample mass (fresh mass, $FW_2$), and this same process was performed for each sample leaf until a series of water potentials and fresh masses were obtained. Then, the sample was dried at 70 °C for 48 h in an oven to calculate the dry mass (DW). Leaf relative water content (RWC, %) was calculated as $(FW_2\text{-}DW)/(FW_1\text{-}DW) \times 100$. A series of water potentials and corresponding RWC were obtained until $\Psi_{leaf}$ did not fall or rise due to cell damage. $C_{leaf}$ and turgor loss point were obtained using the methods outlined by *Brodribb & Holbrook (2003)*.

$C_{leaf}$ (mmol $m^{-2}$ $MPa^{-1}$) was normalized by leaf area as: $C_{leaf} = \delta RWC/\delta \Psi_{leaf} \times (DW/LA) \times (WW/DW)/M$, where $\delta RWC/\delta \Psi_{leaf}$ ($MPa^{-1}$) is the slopes of the curves obtained, DW is leaf dry mass (g), LA is leaf area ($m^2$), WW is mass of leaf water content at 100% RWC (g), and M is molar mass of water (g $mol^{-1}$; *Brodribb & Holbrook, 2003*).

$K_{leaf}$ was calculated using the equation: $K_{leaf} = C_{leaf} \ln(\Psi_0/\Psi_f)/t$, where $\Psi_0$ is the initial leaf water potential of the samples just collected at dawn, and $\Psi_f$ is the leaf water potential after rehydration for t seconds (*Brodribb & Holbrook, 2003*). For percentage loss of leaf hydraulic conductance measurement, branches were cut at dawn when $\Psi_{leaf}$ was high and wrapped in a black plastic bag to balance the water potential. Two compound leaves were detached to measure $\Psi_0$, then two adjacent compound leaves were detached for rehydration with their petioles under water for 15–30 s to obtain $\Psi_f$. $K_{leaf}$ was calculated using the above equation. The average value of $K_{leaf}$ with $\Psi_0 \geq 0.5$ MPa was defined as the maximum $K_{leaf}$ ($K_{lmax}$), thus the leaf relative embolism (%) was calculated as $(1-K_{leaf}/K_{lmax}) \times 100$.

## Water content and sap osmotic potential

The remaining branches for the PLC measurements were also used to measure water content. A 5–6 cm segment and a compound leaf from the middle of each branch were cut, respectively ($n = 6$). After sample fresh masses (FW) were measured, the samples were dried at 80 °C for two days and then weighed (DW). Water content (%) was calculated as $(FW/DW - 1) \times 100$ (*Charrier, Cochard & Améglio, 2013a*).

Xylem sap was extracted from fresh branch samples by centrifugation. The middle parts of the branches were cut to short segments and placed in a 10 ml centrifuge tube. The sap was collected by centrifugation at 8,000 rpm for 10 min (Eppendorf 5804R, Germany). The segments were taken out and discarded, then the precipitates were removed by centrifugation at 10,000 rpm for one minute. The supernatant was used for osmotic pressure measurement via an osmometer (Osmomat 030, Gonotec, Berlin, Germany). The osmotic pressure was then converted into osmotic potential ($\Psi_s$) as: $\Psi_s = -iCRT$, where, '$-i$' represents the dissociation coefficient of solute, 'C' represents the mass molar concentration (*i.e.*, measured osmotic pressure), 'R' represents the gas constant and 'T' represents the thermodynamic temperature.

## Root pressure and soil temperature

Xylem pressure was measured on branches close to the ground (*Améglio et al., 2003*). The top of the branch was cut off under water and discarded, and the cut surface of the remaining branch was trimmed using a fresh razor blade. The cut surface was tightly connected to a hard-wall silicon tube filled with degassed distilled water, then a pressure transducer (CX136-4, OMEGA, Reedville, VA, USA) was connected to the other end of the silicon tube. A data logger (CR1000, Campbell, Logan, UT, USA) was used to record xylem pressure at 15-min intervals for at least 24 h. Six branches from 1–3 individual plants were measured simultaneously. The maximum pressure value was obtained by averaging the six maximum xylem pressures.

At the same time, soil temperature was measured 15 cm below the ground surface with a geothermometer (a common thermometer used to measure soil temperature). This depth was chosen because most roots are distributed between 15 and 20 cm below the surface. Since the diurnal variation of soil temperature was small, it was recorded once a day and the average value of the soil temperatures on the day root pressure was measured, and 2–3 days before and after that day, was taken as the soil temperature for that time.

## Electrolyte leakage

Electrolyte leakage (EL) is often used to test frost damage to living cells (*Charrier, Cochard & Améglio, 2013a*; *Fernández-Pérez et al., 2018*). We measured branch and leaf EL according to the methods of *Fernández-Pérez et al. (2018)*. A 15 cm segment and 3–4 compound leaves from the middle of each branch were cut, respectively. Fresh branch and leaf samples were cut into small pieces, then washed in distilled water and placed in a test tube immersed in deionized water at room temperature (20 °C) for 24 h. The test tube was gently shaken by an oscillator. The electrical conductivity ($C_i$) of the samples was measured using a conductivity meter (STARTER 3100C, Ohaus, Parsippany, NJ, USA). The samples were then autoclaved at 120 kg cm$^2$ for 10 min. After cooling at room temperature, the samples were remeasured for electrical conductivity ($C_f$). EL (%) was calculated as ($C_i/C_f$) $\times 100$.

## Branch nonstructural carbohydrate content

Nonstructural carbohydrates mainly include soluble sugars (SS) and starch (*Dietze et al., 2014*), so we measured the SS and St content of the branch samples. A 15 cm segment was cut from the middle of each branch and used for measurements. The samples were immediately put into an oven at 105 °C for 15 min to stop all enzyme activity, then dried at 70 °C for 48 h to constant mass. The dried samples were then crushed and used to measure NSC content using a modified anthrone method (*Mitchell et al., 2013*). A 0.1 g crushed sample was put in a 10 ml centrifuge tube filled with 5 ml 80% alcohol (volume ratio), then incubated in an 80 °C water bath for 30 min with continuous artificial shaking and mixing. After the mixture was centrifuged at 3,500 rpm for 10 min, the supernatant was transferred to another 10 ml centrifuge tube. Centrifugation and transferring of the supernatant were repeated twice with 80% alcohol. Then, 10 mg of activated carbon was added to the supernatant to decolour it for 30 min. The supernatant was then filtered through a funnel into a 100 ml volumetric flask to measure SS content. The volumetric

flask was filled with distilled water to 100 ml, and 2 ml of the SS solution was placed in a 10 ml centrifuge tube for an anthrone-sulfuric acid reaction in a boiling water bath for 10 min. After cooling, the absorption value at 620 nm was measured by a spectrophotometer (U2910, HITACHI, Tokyo, Japan) to determine the concentration of SS.

The remaining precipitate was used to measure starch concentration by adding 2 ml of distilled water and placing it in a boiling water bath for 15 min. After cooling in ice, 2 ml of 9.2 mol $L^{-1}$ perchloric acid was added to degrade the starch into soluble sugar for 15 min. The mixture was centrifuged at 4,000 rpm for 10 min, and the supernatant was transferred to a 100 ml volumetric flask. The precipitate was re-degraded by 2 ml of 4.6 mol $L^{-1}$ perchloric acid for 15 min, and the supernatant was transferred into the volumetric flask. The precipitate was then washed 2–3 times with 5–6 ml distilled water. The pooled supernatants were considered SS and determined by photometric method, as stated above. The concentration of starch was obtained by dividing the SS concentration by the converting coefficient of glucose into starch.

### Leaf net photosynthetic rate

Leaf net photosynthetic rate was measured in the morning of a sunny day (09:00–11:00 h) in the middle of October and in the middle of December, respectively, in order to analyze the influence of frost on photosynthesis in winter (Li-6400, Li COR, Lincoln, USA). Fully-developed, mature leaves were selected for measurement. Ambient $CO_2$ concentration of 410 $\pm$30 $\mu$mol mol$^{-1}$ was used, and the photon flux intensity was set to 1,500 $\mu$mol m$^{-2}$ s$^{-1}$ (The light saturation point for *P. propinqua* is $\leq$1,500 $\mu$mol m$^{-2}$ s$^{-1}$). After the leaf adapted to the chamber environment for 1–2 min and the stomatal conductance reading was stable, the measurement was recorded.

### Statistical analyses

All values were calculated as mean $\pm$ standard error (SE; $n = 6$). A one-way ANOVA was used to test differences of the indexes measured, with $\alpha = 0.05$ indicating a significant difference. The statistical analyses were all performed with SAS (SAS Institute Inc., Cary, NC, USA).

## RESULTS

### Water potential and embolism changes

Branch water potential decreased from −0.55 MPa in October to a minimum of −3.5 MPa in January, then rose again to −0.6 MPa in June (Fig. 2). Branch native embolism showed the opposite trend, with an increase from 20–30% in October to about 60% in January with a slight decrease in December, then gradually decreasing back to between 20% and 30% in June (Fig. 2).

Leaf embolism also increased progressively during winter, reaching a maximum in January near 90%, then returning to the original level of about 20% (Fig. 2). Compared to branch native embolism, leaf embolism was more pronounced.

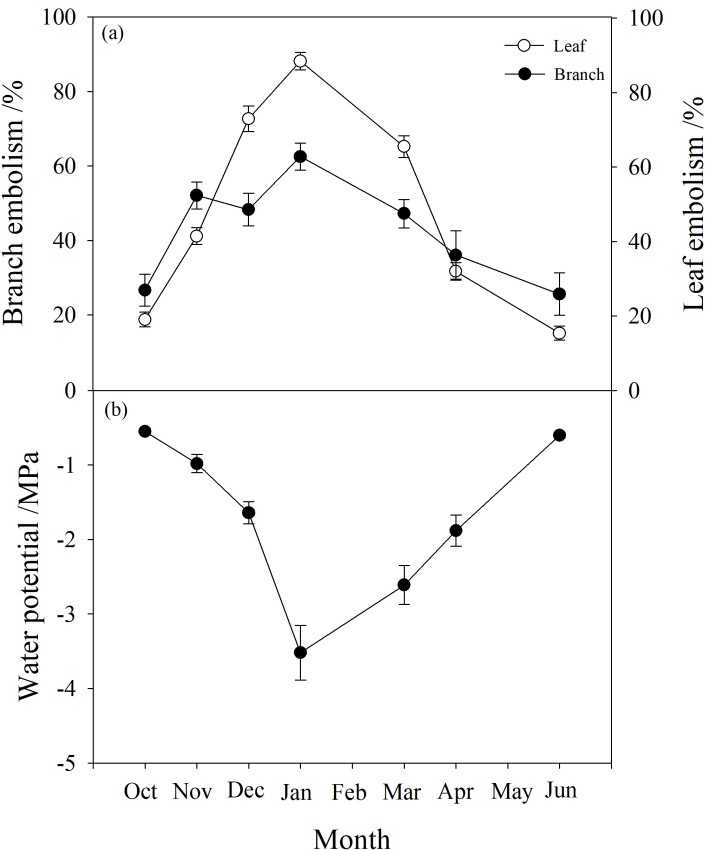

**Figure 2** **Seasonal changes in percentage loss of conductivity (PLC) and water potential of *P. propinqua* plants.** (A) Predawn PLC of branch and leaf; (B) Predawn twig water potential. Values are means ± standard errors ($n = 6$).

## Branch and leaf water content

The water content of branches and leaves showed an opposite trend. In winter, the water content of leaves gradually decreased to below 40%, while that of branches gradually increased to 80%. However, in autumn and spring (October, May and June), the water content of leaves was higher than that of branches (Fig. 3).

## Osmotic potential of branch xylem sap

The osmotic potential of xylem sap showed a downward trend from autumn to winter, fluctuating between −3 and −2 kPa during winter. It continued to decline to a minimum below −3 kPa in March, then rose to the level of the previous autumn, between −1.5 and −1 kPa (Fig. 4).

## Root pressure changes along with soil temperature

*P. propinqua* had a maximum root pressure of 28 kPa in autumn, and in November it still had a positive pressure of 4.6 kPa when soil temperature was 9 °C. During the months of December, January and February when the soil froze, there was no positive pressure

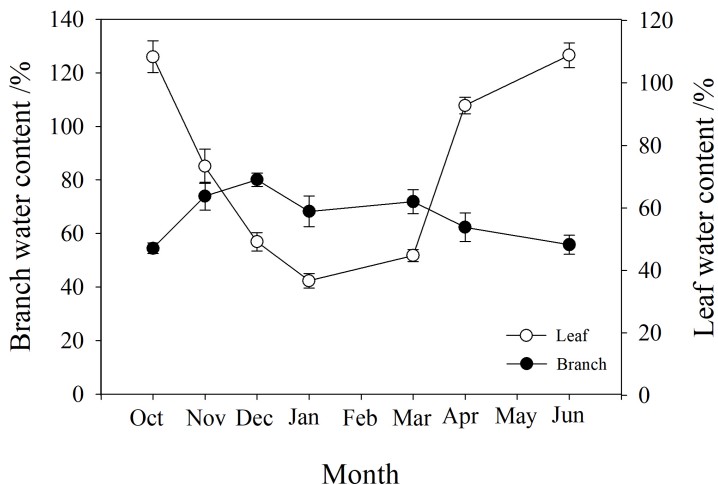

**Figure 3 Seasonal changes in branch and leaf water content of *P. propinqua* plants.** Values are means ± standard errors ($n = 6$).

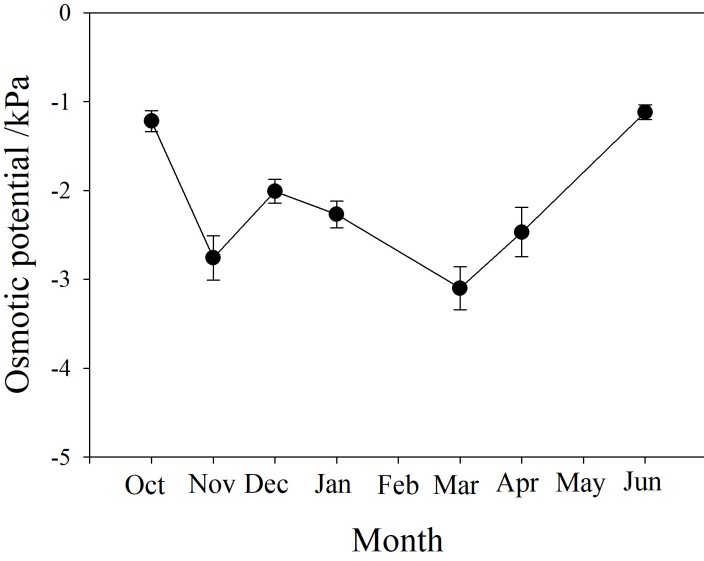

**Figure 4 Seasonal changes in xylem sap osmotic potential of *P. propinqua* plants.** Values are means ± standard errors ($n = 6$).

detected. Positive root pressures appeared again along with the increase in soil temperature from $-6\,°C$ in January to $11\,°C$ in March, reaching a maximum of 20 kPa in April (Fig. 5).

## Branch and leaf electrolyte leakage

Branch and leaf electrolyte leakage (EL) increased from autumn (October) to winter (January) and decreased in the spring (from March to June). However, branch EL was significantly higher than leaf EL at all time points ($P < 0.05$). In January, branch EL reached a maximum of 38.7%, and leaf EL was only 23.1% (Fig. 6).
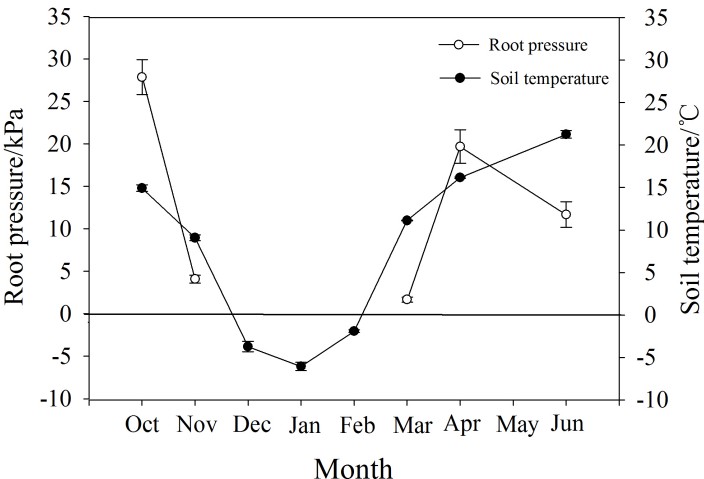

**Figure 5** Changes in root pressure in relation to soil temperature in *P. propinqua* plants. Values are means ± standard errors ($n = 6$).

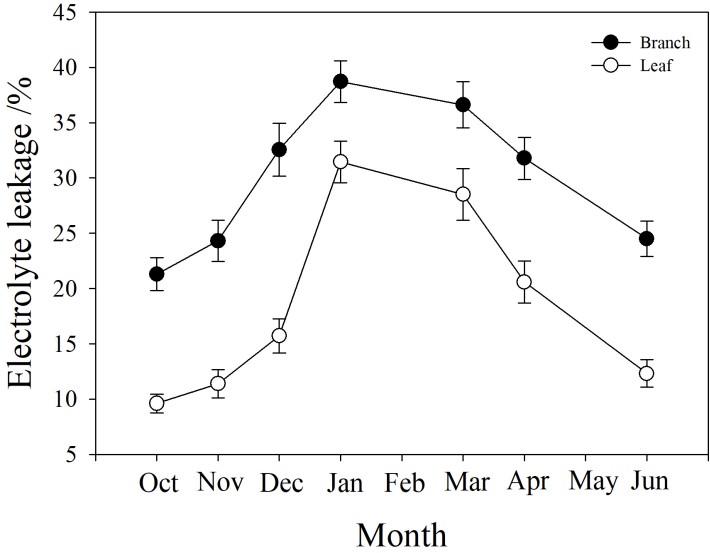

**Figure 6** Seasonal changes in branch and leaf electrolyte leakage in *P. propinqua* plants. Values are means ± standard errors ($n = 6$).

## Net photosynthetic rate and NSC content

Leaf net photosynthetic rate, stomatal conductance and transpiration rate in summer were all significantly higher than that in winter. In winter, *P. propinqua* still had a net photosynthetic rate of $4.20 \pm 0.33\ \mu\text{mol CO}_2\ \text{m}^{-2}\ \text{s}^{-1}$ (Table 1).

Soluble sugar (SS) content in the branch increased from October to March, peaking at near 12%, and then decreased. Starch (St) content decreased slightly from October to December, and then increased to a maximum in January near 8%, followed by a reduction from March to June (Fig. 7).

**Table 1  Net photosynthetic rate, stomatal conductance and transpiration rate of *P. propinqua* plants.** Values are means ± standard errors ($n = 6$), and different letters refer to significant differences at $P <$ 0.05.

| | Net photosynthetic rate ($\mu$mol $CO_2$ m$^{-2}$ s$^{-1}$) | Stomatal conductance (mmol $H_2O$ m$^{-2}$ s$^{-1}$) | Transpiration rate (mmol $H_2O$ m$^{-2}$ s$^{-1}$) |
|---|---|---|---|
| Autumn | 9.30 ± 1.10 a | 70.22 ± 8.16 a | 1.30 ± 0.20 a |
| Winter | 4.20 ± 0.33 b | 33.75 ± 4.08 b | 0.40 ± 0.04 b |

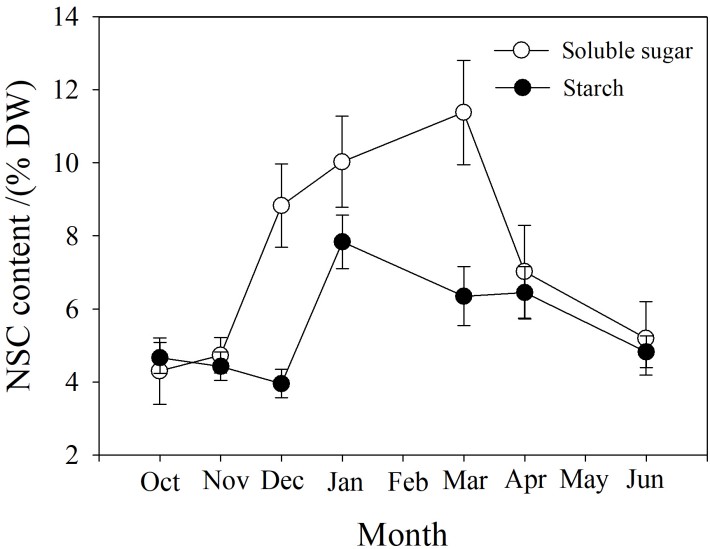

**Figure 7  Seasonal changes in soluble sugar and starch content in *P. propinqua* branches.** Values are means ± standard errors ($n = 6$).

# DISCUSSION

The mean vessel diameters of the culm and branch of *P. propinqua* were 68.95 ± 1.27 $\mu$m and 33.97 ± 0.89 $\mu$m, respectively (Table 2), wider than most diffuse- and ring-porous species (20–40 $\mu$m; *Hacke et al., 2017*), expected to be more vulnerable to severe winter embolism. However, *P. propinqua* experienced a relatively mild winter embolism compared to other temperate diffuse- and ring-porous species, which experience very severe winter embolisms, with degree of xylem embolism near to 100% (*Sperry et al., 1994; Cochard et al., 2001; Jaquish & Ewers, 2001; Christensen-Dalsgaard & Tyree, 2014*). Based on our previous studies, compared with diffuse- and ring-porous species growing in the same area (*Dai, Wang & Wan, 2020*), *P. propinqua* experienced less severe winter embolisms, showing a weak trade-off between vessel diameter and freeze-thaw-induced embolism severity. *P. propinqua* may have some strategies, such as embolism avoidance or embolism repair, to prevent or cope with freeze-thaw-induced embolism.

**Table 2  Vessel diameter in different parts of *P. propinqua* plants.** Values are means ± standard errors ($n = 6$).

| Culm (μm) | Branchlet (μm) | Petiole (μm) | Midrib of leaves (μm) |
|---|---|---|---|
| 68.95 ± 1.27 | 33.97 ± 0.89 | 9.68 ± 0.22 | 7.21 ± 0.16 |

## The role of root pressure in stem resistance to freeze-thaw-induced embolism

Root pressure was first identified by Hales in 1727, but the function and mechanism of root pressure have long been questioned (*White, 1938*; *Schenk, Jansen & Hölttä, 2021*; *Drobnitch et al., 2021*). Previous studies have focused on this area of research, and a consensus has been reached on the function of root pressure in water transport, especially in embolism repair (*Priestley, 1920*; *O'Leary, 1966*; *Pickard, 2003a*; *Pickard, 2003b*; *De Swaef et al., 2013*; *Gleason et al., 2017*). Our studies showed that the reduction of branch native embolism in spring was accompanied by the occurrence of root pressure beginning in March, indicating the positive effect of root pressure on seasonal embolism repair. These results are consistent with studies on the recovery of hydraulic conductivity of other woody plants in spring (*Sperry et al., 1994*; *Hacke & Sauter, 1996*; *Améglio et al., 2003*; *Cobb, Choat & Holbrook, 2007*). According to Henry's law, the critical pressure required to dissolve bubbles is a negative value, meaning an embolism can be refilled when positive pressure exists in the xylem (*Yang & Tyree, 1992*).

Root pressure may also affect the severity of winter embolism due to timely embolism repair. Several researchers have demonstrated that species that generate xylem pressure exhibit relatively mild winter embolisms, showing a weak trade-off between the degree of freeze-thaw-induced embolism and vessel diameter (*Améglio et al., 2004*; *Ameglio et al., 2006*; *Niu, Meinzer & Hao, 2017*). This positive xylem pressure has always been considered stem pressure. The generation of stem pressure is related to the xylem sap osmotic potential established by the sugar transport from xylem parenchyma cells to vessels, activated by freeze-thaw cycles (*Améglio et al., 2004*; *Cirelli, Jagels & Tyree, 2008*). The osmotic potential of vessels increases with stem pressure and the embolism is repaired, thus, stem pressure-generating species may not suffer from serious embolism in winter. Root pressure is functionally different from stem pressure and is considered to be osmotic in origin due to the osmotic potential gradient generated by the active absorption of solutes from the soil by the roots; therefore, root pressure is related to root activity (*Oosterhuis & Wiebe, 1986*; *Priestley, 1920*; *Ewers et al., 2001*). Root activity is largely affected by soil temperature, and for most temperate woody plants, 6 °C is a critical temperature for root growth, so there are few reports on the detection of root pressure in winter (*O'Leary, 1966*; *Ameglio et al., 2006*). However, for *P. propinqua*, root pressure still occurred in November when soil temperature was 9 °C, which may contribute to embolism repair in this species.

The energy required to produce root pressure comes from respiration, which decreases with temperature (*White et al., 1958*; *Atkin & Tjoelker, 2003*). Low temperature also affects root pressure by reducing the activity of $H^+$-ATPase in the root plasma membrane, as the sensitivity of roots to low temperatures is related to root hydraulic properties

through aquaporins and active proton pumping (*Lee & Chung, 2005*). After exposure to temperatures of 8 °C for one day, H$^+$-ATPase in the plasma membrane of cucumber (*Cucumis sativus* L.) seedlings decreased significantly (*Lee et al., 2004*). However, there are species differences in root sensitivity to low temperatures (*Bloom et al., 2004*). Furthermore, the root system of spinach (*Spinacia oleracea* L.) was found to have the ability to rapidly acclimate to temperature changes and continued to acclimate during extended exposure to low temperatures (5 °C; *Fennell & Markhart, 1998*). Similarly, the roots of *P. propinqua* were well adapted to low temperatures, which may be attributed to long-term adaptation to freezing conditions. It is hypothesized that root pressure helped initial stem resistance to freeze-thaw-induced embolism in these plants and further prevented severe embolism with increasing freeze-thaw cycles. However, the root pressure in November was very low, only 4.6 kPa, which was not sufficient to repair the whole plant, so there may be other mechanisms at play.

## The role of leaf 'hydraulic fuses' in stem resistance to freeze-thaw-induced embolism

It is generally believed that there is 'vulnerability segmentation' among the root-stem-leaf hydraulic transport system to drought stress in many plant species, or that different organs have different vulnerability to embolisms (*Liu et al., 2020*). These plants always limit hydraulic failure to easily-replaced organs, usually leaves, to protect the hydraulic integrity of the whole plant, like a 'hydraulic fuse' (*Zufferey et al., 2011*; *Zhang, Zhang & Cao, 2017*). Previous studies have shown that petioles are more sensitive to drought-induced embolism than stems (*Tyree et al., 1993*; *Tsuda & Tyree, 1997*; *Zhu et al., 2016*). Another study demonstrated that bamboo leaves have the 'hydraulic fuse' function, because of a very thin, waxy leaf cuticle resulting in easy water loss (*Yang et al., 2012*). Our results showed that leaves suffered more severe winter embolisms than the branches, likely because of their significant desiccation, reflected by the low water content of leaves in winter. *Di Francescantonio et al. (2020)* reported that evergreen species were more cold-tolerant than deciduous species because they had hydraulic segmentation, and a significant loss of leaf hydraulic conductance could reduce freezable water in tissues, thus delaying freezing under cold conditions. The branches of *P. propinqua* exhibited higher water content than the leaves, so it unlikely that the leaves could utilize the water stored in the stem for transpiration.

It was notable that the leaves suffered less frost damage to living cells than the stems, because the leaves had lower electrolyte leakage (EL) than the stem. This indicated that the cell membranes in the leaves were not damaged by freezing, or that apoplastic freezing did not induce symplastic freezing. When ice seeding originates in the stem and spreads rapidly to leaves, the leaves may have some mechanisms to avoid freezing, including a barrier between the stem and the leaf to prevent ice propagation. This barrier may be made up of rigid cell walls, which keep the cell wall in tight contact with the cell membrane to prevent ice seeding in the space between them, and that do not easily change shape with extracellular ice, posing less physical damage to cell membranes (*Zhang et al., 2016*). Moreover, the leaves of many species, including bamboo, can utilize supercooling (without

freezing) as a cold tolerance strategy to survive subzero temperatures (*Pearce, 2001*). These attributes of the leaves help evergreen plants to survive winter.

## The role of cold acclimation in stem resistance to freeze-thaw-induced embolism

Under natural conditions, plants undergo cold acclimation with the gradual decrease of air temperature from autumn to winter (*Arias et al., 2017*). Cold acclimation initiates a series of physiological and biochemical adjustments to improve the freezing tolerance of plants, including the accumulation of soluble sugars, a decrease in osmotic potential, an increase in the amount of cryoprotectant substances, and cell dehydration (*Ameglio et al., 2006*; *Zhang et al., 2016*; *Arias et al., 2017*; *Fernández-Pérez et al., 2018*). We found that the water content of branches increased in winter, peaked in early spring in March, with a relatively high water content in winter, consistent with the results of *Lintunen et al. (2018)* who reported that branch water content in winter was slightly lower than in early spring. Many studies have found that frost resistance is related to the decrease of water content in the early stage of cold acclimation, likely because dehydration can protect against frost by slowing growth or concentrating soluble carbohydrates and osmoregulation substances such as proline (*Sperling et al., 2017*; *Baffoin et al., 2021*). *Lintunen et al. (2018)* also found that branches with lower relative water content froze at lower temperatures. Unexpectedly, branch water content was not the lowest when the embolism was at its maximum, showing no direct correlation between them. This may be due to the absorption of water by roots on sunny days during the winter, or due to the transfer of water from the apoplast to the symplast during thawing (*Sparks, Campbell & Black, 2001*; *Charra-Vaskou et al., 2016*). The water was likely stored in the symplast spaces in order to prevent cell death due to dehydration, or to be used for embolism repair or spring growth.

Soluble sugar content is an important parameter during cold acclimation. Previous studies show that soluble sugar (SS) content in the tissue of plants growing in temperate and cold regions peaks in midwinter. This late peak in SS content improves the frost resistance of these plants because SS protects the plant through osmosis, improving the fluidity and stability of the cell membrane, preventing freezing damage of the tissues (*Charrier et al., 2013b*; *Fernández-Pérez et al., 2018*). Previous studies have found that frost resistance is closely related to soluble sugar content (*Morin et al., 2007*; *Baffoin et al., 2021*). Moreover, tissue water content often decreases as SS content increases, indicating a synergistic effect of the two factors (*Charrier & Améglio, 2011*; *Charrier et al., 2013b*). SS often comes from the conversion of stored starch (St) to sugar in winter (*Ameglio et al., 2006*). In our study, SS content was also higher in winter and peaked in March. St content decreased as SS content increased, but St content was actually highest in January, which may be because *P. propinqua* is an evergreen species that can photosynthesize in winter. There is also relatively less demand for carbon in winter, so it is stored as St.

In winter, xylem sap osmotic potential decreased because of water content decreases in vessels and the accumulation of osmolytes, which consist of sugars, other soluble compounds and ions (*Schill et al., 1996*; *Ewers et al., 2001*; *Westhoff et al., 2008*). We observed that the xylem sap osmotic potential of *P. propinqua* was lower in winter, with

some fluctuations, and reached its lowest point in March, almost opposite to the change trend of SS during this period, but not completely. This suggests that SS is not the only osmotic substance, and that other substances also played a role as osmosis substances. Solutes can reduce the freezing point (1–2 °C; *Pearce, 2001*), which does not help stop freezing, but can reduce the number of freeze-thaw cycles and freeze-thaw-induced embolism severity (*Lintunen et al., 2018*). Therefore, water content, SS content and xylem sap osmotic potential can be used in combination to predict the cold acclimation process of *P. propinqua*, which the plant uses to survive cold conditions. Cold acclimation, as a survival strategy, can improve the freeze resistance capabilities of both temperate tree species and tropical tree species (*Mayr & Ameglio, 2016*; *Ayala-Jacobo, Woeste & Jacobs, 2021*).

## CONCLUSIONS

It has been widely accepted that there is a strong trade-off between vessel diameter and risk and severity of freeze-thaw-induced embolism. However, wide-vesseled *P. propinqua* can survive in cold regions with less severe winter embolisms than most temperate broadleaved trees. We suggest that some strategies the plant uses to protect stem hydraulic integrity or prevent cell freezing may contribute to its resistance to freeze-thaw-induced embolisms. *P. propinqua* had root pressure in autumn lasting until November, and its leaves experienced more severe winter embolisms than the branches. Moreover, branch water content, xylem sap osmotic potential and branch NSC content all changed in winter as *P. propinqua* acclimated to the cold in order to adapt to freezing conditions. These results suggest that *P. propinqua* depends on root pressure, the leaf 'hydraulic fuse' function and the cold acclimation process to cope with freeze-thaw-induced embolisms. This study provides insights into the mechanisms by which evergreen plants maintain xylem hydraulic function in cold regions.

## ACKNOWLEDGEMENTS

We greatly acknowledge Wenxin Liu for her assistance with material samplings, and the two anonymous reviewers for their useful comments.

### Funding

This study has been supported by grants from Outstanding Doctoral Award Project of Shanxi Province (SXYBKY201732), and a Shanxi Natural Science Foundation Project (201801D121246). The funders had no role in study design, data collection and analysis, decision to publish, or preparation of the manuscript.

### Grant Disclosures

The following grant information was disclosed by the authors:
Outstanding Doctoral Award Project of Shanxi Province: SXYBKY201732.
Shanxi Natural Science Foundation Project: 201801D121246.

## Competing Interests

The authors declare that there are no competing interests.

## Author Contributions

- Yongxin Dai performed the experiments, analyzed the data, prepared figures and/or tables, authored or reviewed drafts of the article, and approved the final draft.
- Lin Wang performed the experiments, analyzed the data, authored or reviewed drafts of the article, and approved the final draft.
- Xianchong Wan conceived and designed the experiments, authored or reviewed drafts of the article, and approved the final draft.

## Data Deposition

The raw data are available in the Supplemental File.

## Supplemental Information

Supplemental information for this article can be found online at http://dx.doi.org/10.7717/peerj.15979#supplemental-information.

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
