# Peer review of "Maintenance of xylem hydraulic function during winter in the woody bamboo Phyllostachys propinqua McClure"

_PeerJ, doi:10.7717/peerj.15979_

## Round 0.1 · original submission · Major Revisions

Dear authors,

I have now received two reviews for your manuscript. Both found that your results are of interest but that the manuscript need substantial improvement before it can be reassessed. I concur with their assessment and you will find below my own comments and suggestions regarding your manuscript in addition to what the reviewers pointed out. I thus encourage you to submit a revised manuscript with a point-by-point answer to the reviews. Please note that the revised manuscript will be reassessed and I cannot guarantee that it will be accepted. I look forward reading the revised manuscript.


Comments:

The most critical issues for me is the lack of information about what those plants actually experienced and thus how can the results be interpreted. What were the environmental conditions during the experiment? Could you provide a plot of temperature, etc.? Is it likely to have a large impact on the results and might need to be included in the analyses as a covariate. This is important to assess when there was freezing (EL, etc.) in relation to the measurements. Temperature will also impact other processes (sugar to starch conversion, etc.). Thus, some assessment of the environmental conditions effect of the measured variables is important.

Additionally, a big part of the discussion is very general (more like providing information that could have been in the introduction). The manuscript would benefit from

As pointed it by the reviewers, the English needs to be improved. Asking a fluent speaker or a professional service to review the manuscript and edit the language before submission would be necessary.

Additionally, I wonder if you could go further with your analyses by for example looking how the different variables of interest might be correlated and if that might tell you something about the plant response.

L59 see Lintunen, A., Salmon, Y., Hölttä, T. & Suhonen, H. (2022) Inspection of gas bubbles in frozen Betula pendula xylem with micro-CT: Conduit size, water status and bark permeability affect bubble characteristics. Physiologia Plantarum, 174( 4), e13749. Available from: https://doi.org/10.1111/ppl.13749

L96 I don’t think you can claim that. NSC are used as building block and energy source mainly. Furthermore, osmoregulation involved other compounds than NSC.

L114, no building new conduits is another way

L136 natural conditions meaning, in soil, and under non controlled climatic conditions?

L141 and around: experimental design unclear. How plants were used for each sampling point (I.e. Is one plant used for the whole set of measurement, or different plants?). Please also indicate here that n=6.

L166 more details are need to explain how you dealt with the vessel length issue. It is unclear for me how these 3-4cm segment would allow to do that. Also the hollow stem issue

L221 why use water here and not follow Ameglio’s protocol? Was there instances in which the water froze?

L226 how often was the soil temperature recorded? It is unclear from Fig. 4 is the point is an average or a unique value.

L266 is CO2 and light are set, you are not measuring the actual photosynthesis but some quite of maximum photosynthesis or photosynthetic capacity. How about the temperature in the cuvette and for the plants, was it controlled? And the humidity?

L292 here for example we need to see soil temperature at the time of root pressure measurement

L301 here again without VPD, etc. it is difficult to understand what the gas exchange data mean

L312 but how much freezing conditions did they really face?

L329-355 this is more introduction material than discussion since there is nothing about the results. Please

L357 what is lethal freezing temperature for this species?

L380 what would be the barrier?

Reviewer 1 ·

Basic reporting

This study provides interesting data on xylem hydraulic function during the winter season in bamboo. The paper is generally well written, but there are some questions and areas that need to be improved. Please address them below.

Experimental design

I have nothing to add.

Validity of the findings

I have nothing to add.

Additional comments

About Terms
1. You use the terms freezing-induced embolism and freeze-thaw-induced embolism. My understanding is that most researchers use freeze-thaw-induced embolism or freeze-thaw cycle-induced embolism. Why not unify them into one or the other?
2. you seem to be using several terms, branch, culm, twig and stem, in a confusing manner to describe the measuring material. Can they be unified under one of them? Or do they each have their own meaning?

About References
You have cited the appropriate references in this paper. The following paper would also be useful for your research. Please consider them.
Taneda H, Tateno M. 2005. Hydraulic conductivity, photosynthesis and leaf water balance in six evergreen woody species from fall to winter. Tree Physiology 25: 299-306.

Materials and methods
Line 135
Please describe the frequency of irrigation and the amount of irrigation with numerical values.

Line 139-140
Please describe the source of the meteorological data citation.

I cannot image the size of the bamboo used in the experiment. Please describe the tree height and the stem diameter. Please describe from which position on the bamboo you collected the leaves and branches.

Line 159 optical microscope
Please describe the company name of optical microscope and its product number, etc.

Line 171-172Bamboo trunks are hollow; 3 -4 cm segments with node may also contain hollow sections. I can't imagine how to measure the conductivity.

Line 197
initial → rehydration ?

Line 209
Centrifuge
Please describe the company name of centrifuge and its product number, etc.

Line 226
geothermometer
Please describe the company name of geothermometer and its product number, etc.

Line 233
What is the temperature of the water in a water bass?

Line 234  Conductivity meter
Please describe the company name of conductivity meter and its product number, etc.

Line 250 – 251
spectrophotometer
Please describe the company name of spectrophotometer and its product number, etc.

Line 270 – 271
You present your data as means ± SD and test for their significance with ANOVA. As I understand it, ANOVA is a method to test the difference between the population means. Therefore, shouldn't it be SE (standard error) instead of SD?

Line 288 – 292
Please correct kpa, the unit of pressure, to kPa. The p in Pascal is capitalized.

Line 311
Why don't you describe the values of vascular diameter of diffuse- and ring-porous species for comparison with bamboo, citing papers or books? It would help readers understand better.

“Frontiers in Forests and Global Change” is in italics.

Line 566 – 5567
In Progress in Botany 77 (pp. 381-414). Springer Cham. → In : author name Progress in Botany. Springer, 381-414.

Line 596
Plant physiology → Plant Physiology

The paper discusses the relationship between various measurements and temperature. I have one suggestion. How about adding a chart that shows the change in temperature (maximum, minimum, and average) during the experiment? I think this will help readers understand better.

Figure 1
cavitation embolism → percent loss of hydraulic conductivity (PLC)
Stem embolism → PLC of stem, Leaf embolism → PLC of leaf

Figure 3
Vertical axis kPa → kPa

Figure 4
Vertical axix kPa → kPa

Reviewer 2 ·

Basic reporting

The authors present a nice and sound set of data on strategies of a bamboo species to confront freezing-induced embolism. I recommend this manuscript for publication, but I strongly suggest a detailed revision and consideration of all points raised in my evaluation.

My main concerns are:

English language absolutely needs to be improved. I have made many annotations on the manuscript in which a thorough revision is needed. I have also suggested many corrections on the text. So, in general, this revision must be done. I suggest the authors find a fluent English speaker or contact an editing service.

Throughout the introduction, the authors mention, and cite references which correspond to 'trees' (e.g. L37, L39, L48, L69, L89) and not 'plants' or 'woody plants'. Reading these beginning paragraphs gives the sensation that the authors will present results for a tree species and then in L109 they present Bamboos, 'woody grass species'. All that is presented in the introduction also applies to all growth-forms, so I would refer to woody plants, so that the emphasis on trees is reduced.

L132. Can the authors determine 'frost hardiness' from the methods and results presented? Is this really one of the objectives of the study? I feel they are determining 'hardening' of the xylem transport system. They are not determining direct effects of freezing or freezing temperatures on survival of this species. I believe that for frost hardening determinations, they would need to determine temperatures at which injury occurs.

What do the authors mean by normal watering management (L136). This is very important because the water relations of this species is strongly dependent on soil conditions. The only information the reader has is the mean annual precipitation, but were these plants irrigated throughout the experiment? Were soil conditions monitored during the measurement period? Are differences in the predawn water potentials solely determined by osmotic adjustments or did soil water potential conditions vary through the experiment?

L435. How can the authors be sure P. propinqua responds through cold acclimation as a strategy to resist freezing temperatures in winter? Frost resistance was not measured, i.e. injury temperautres were not determined. So, could it be possible that frost resistance is nearly constant all year round, in other words, injury temperatures are similar throughout the year, and that increases in SS (and other substances) only play a role in the water relations of the plant, mainly for repairing embolism?

L437. Their results actually provide insight into the mechanisms by which evergreen plants may deal with xylem hydraulic function... but not with respect to frost resistance... their focus was on hydraulic function, not frost resistance.

Many other suggestions and corrections are detailed on the attached manuscript.

Experimental design

Minor comments on methods are detailed on the manuscript.

Validity of the findings

no comments

Annotated reviews are not available for download in order to protect the identity of reviewers who chose to remain anonymous.

---

## Round 0.2 · Minor Revisions

Dear authors,

Thank you for submitting your revised manuscript to PeerJ. I have received two reviews that both highlight the improvement in the manuscript, but suggest some minor changes to further improve the manuscript. I concur with their assessment and added some further comments and suggestions below.

Best regards

L12 cavitation or embolism, choose one-year

L29 increased from what to what?

L41 maintenance of hydraulic function already in the title, no need to add as a key work

L48 in case of symplastic freezing, this not the sap that is freezing (except for the phloem sap)

L52 it depends a lot if the plants are winter hardened or not. If there are then usually apoplastic freezing is not accompanied by symplastic freezing, but if not, then symplastic freezing will likely occur at the same time

L67 dissolved

L69 “an embolism” delete “an”

L107 there are only very few example of embolism repair under tension

L117-120 not really relevant

L187 do you mean that this species are growing at the same sites at the studied bamboo?

L225 do you mean: When the top 5cm of the soil appeared dry

L237 a sampling interval of one to two months

L262 the given vessel length (longer than the whole trunk >2 or 3 m) seems really long compared to other reported studies I know of bamboo (20 - 100cm)

L295-297 is that really sufficient time and pressure to restore full conductivity? Pressure is the same or smaller than the one used for native conductivity. An insufficient rehydration would provide you with an underestimated Kmax value, meaning that you would underestimate the PLC. While this is unlikely not affect the conclusion, it might (it the problem exist) call for name reflecting that situation ("estimated PLC" or "relative PLC")


L307 sample mass not weight (weight is a force scientifically speaking)

L328 is that sufficient for rehydration? See comment L295-297

L354 Campbell

L381 instead of always mentioning the same sampling as PLC, why not have a short section (before any measurements method) describing the sampling strategy and method and then state that those were applied for all following methods

L451 specify how you treated the cases in which more than one measurement was taken per sampled plant (e.g predawn water potential, for which n=4-5, no?)

L470 % on a mass based?

L476 “making it more vulnerable to severe winter embolism”. I would say that it could be expected to be more vulnerable, since the next sentence say it is not

L486 why “may have some”? That would fit in the intro, but here you have at least some element of response. Rephrase accordingly

L564 while there are evidence of hydraulic segmentation in a number of species, it is certainly not a generally accepted property across species-specific

L581 utilize for what?

L65 do you know anything about that species cold hardiness ability? For example, production of antifreeze protein etc

L659 less severe than what?

L674 please acknowledge the work of the two anonymous reviewers who helped improved the manuscript through their comments

Reviewer 1 ·

Basic reporting

I have nothing to add.

Experimental design

I have nothing to add.

Validity of the findings

I have nothing to add.

Additional comments

The description of the data in the tables and figures is not sufficient. Please add the following sentences.

Values are means ± standard errors (n = ?), and different letters refer to significant difference at P < 0.05.

Reviewer 2 ·

Basic reporting

Authors have done a nice job in their consideration of all my questions, comments and suggestions. I have added a few comments in this new version of the manuscript.

Comments and suggestions:
P1 L18. electrolyte
P2 L42. Sap freezing within the plants may occur in two different ways...
P3 L88. Species that are capable of producing root pressure may expand their distribution ranges (Ewers...
P4 L91. Celastrus orbiculatus in italics
P5 L145. Do not refer to ´tree-like woody bamboo with a trunk...´ Why not refer to ´clumping bamboo with culms that reach 2-3 m in height...
P6 L159. What is normal growth?
P6 L160-168. One very long sentence...
P6 L179. ... entire culm. I would delete ´trunk´ from the text and include ´culm´ hereafter...
P7 L184. Delete ´that can take photos´. Maybe something like ...equipped with a XXX camera.
P7 L197 and L201 and hereafter... delete underwater and replace with under water.
P8 L229. You have defined leaf water potential as Ψl (L211), should be consistent throughout the manuscript, here you define leaf water potential as Ψf. In the same manner, does Ψo correspond to the predawn leaf water potential (Ψpd)? If so, be consistent.
P8 L238. Delete ´Branch and leaf water content was then measured.´
P8 L248 and L249. Delete rmp and replace with rpm.
P10 L281. cm2 (superscript).
P12 L353. CO2 (subscript).
P13 L369. ´Some strategies´... such as? Any suggestions on which strategies?
P13 L376-380. Very long sentence, divide into two sentences. ...embolism repair. These results are consistent with studies on the recovery...
P25 L747. Delete period (.) after branches?

Experimental design

OK

Validity of the findings

OK

---

## Round 0.3 · accepted · Accept

Dear authors, I have now been through the latest revision and assessed that you answered satisfactorily the previous comments from the reviewers and myself. Well done.
Note that while the manuscript is now ready for publication, I noticed the following minor formatting issues. Please correct them:
L77: “ high altitude environments” edit color etc to fit the text
L163 dawn
L233 “potential.Two” a space is missing after the dot.